# The Hemorrhagic Side of Primary Angiitis of the Central Nervous System (PACNS)

**DOI:** 10.3390/biomedicines12020459

**Published:** 2024-02-19

**Authors:** Marialuisa Zedde, Manuela Napoli, Claudio Moratti, Francesca Romana Pezzella, David Julian Seiffge, Georgios Tsivgoulis, Luigi Caputi, Carlo Salvarani, Danilo Toni, Franco Valzania, Rosario Pascarella

**Affiliations:** 1Neurology Unit, Stroke Unit, Azienda Unità Sanitaria Locale-IRCCS di Reggio Emilia, Viale Risorgimento 80, 42123 Reggio Emilia, Italy; 2Neuroradiology Unit, Azienda Unità Sanitaria Locale-IRCCS di Reggio Emilia, Viale Risorgimento 80, 42123 Reggio Emilia, Italy; napoli.manuela@ausl.re.it (M.N.); pascarella.rosario@ausl.re.it (R.P.); 3Stroke Unit, Department of Neuroscience, San Camillo Forlanini Hospital, 00152 Rome, Italy; 4Department of Neurology, Inselspital, Bern University Hospital, University of Bern, 3010 Bern, Switzerland; 5Second Department of Neurology, School of Medicine, Attikon University Hospital, National and Kapodistrian University of Athens, 157 72 Athens, Greece; 6Neurology Unit, Department of Cardio-Cerebrovascular Diseases, Maggiore Hospital ASST-Crema, 26013 Crema, Italy; luigi.caputi@asst-crema.it; 7Rheumatology Unit, Azienda Unità Sanitaria Locale-IRCCS di Reggio Emilia, Viale Risorgimento 80, 42123 Reggio Emilia, Italy; 8Emergency Department Stroke Unit, Policlinico Umberto I, University La Sapienza, 00189 Rome, Italy; danilo.toni@uniroma1.it

**Keywords:** PACNS, ICH, intracranial hemorrhage, stroke, large vessels, small vessels, medium vessels, SVD, CAA, ABRA, CAA-related inflammation

## Abstract

Primary Angiitis of the Central Nervous System (PACNS) is a rare cerebrovascular disease involving the arteries of the leptomeninges, brain and spinal cord. Its diagnosis can be challenging, and the current diagnostic criteria show several limitations. Among the clinical and neuroimaging manifestations of PACNS, intracranial bleeding, particularly intracerebral hemorrhage (ICH), is poorly described in the available literature, and it is considered infrequent. This review aims to summarize the available data addressing this issue with a dedicated focus on the clinical, neuroradiological and neuropathological perspectives. Moreover, the limitations of the actual data and the unanswered questions about hemorrhagic PACNS are addressed from a double point of view (PACNS subtyping and ICH etiology). Fewer than 20% of patients diagnosed as PACNS had an ICH during the course of the disease, and in cases where ICH was reported, it usually did not occur at presentation. As trigger factors, both sympathomimetic drugs and illicit drugs have been proposed, under the hypothesis of an inflammatory response due to vasoconstriction in the distal cerebral arteries. Most neuroradiological descriptions documented a lobar location, and both the large-vessel PACNS (LV-PACNS) and small-vessel PACNS (SV-PACNS) subtypes might be the underlying associated phenotypes. Surprisingly, amyloid beta deposition was not associated with ICH when histopathology was available. Moreover, PACNS is not explicitly included in the etiological classification of spontaneous ICH. This issue has received little attention in the past, and it could be addressed in future prospective studies.

## 1. Introduction

Primary Angiitis of the Central Nervous System (PACNS) is a rare cerebrovascular disease whose pathological hallmark is transmural inflammation of leptomeningeal, cerebral and spinal vessels. The definition, based on pathology, strongly influences the practical possibility of making a definite diagnosis, since the latter requires a sample of the leptomeninges, brain and vessels where the presence of transmural inflammation of the vessel wall is highlighted. The current diagnostic criteria were proposed in 1988 by Calabrese and Mallek [1] and updated in 2009 by Birnbaum and Hellmann [2]. While the first criteria [1] were proposed describing a very small subset of subjects (8) in whom this diagnosis was presumed by using the clinical and diagnostic techniques available in the 1980s, the subsequent update [2] of the criteria refers to the point of view of the authors regarding the differential diagnosis of Reversible Cerebral Vasoconstriction Syndrome (RCVS), then known as Benign Angiitis of the Central Nervous System (BACNS) [3,4], but does not take into account a much wider range of differential diagnosis and a substantial difference in the clinical presentation, natural history and pathophysiology of the two diseases. The clinical manifestations and neuroimaging patterns associated with PACNS can be variable and often non-specific, and they tend to be underreported or reported in an incoherent or non-systematic manner in the available literature, as underlined by the European Stroke Organization guidelines on PACNS [5]. One of the possible patterns of PACNS is intracranial bleeding, in particular spontaneous parenchymal intracerebral hemorrhage (ICH). Its pathophysiology is incompletely understood, and its exact prevalence remains to be determined. In particular, it is not known whether the natural history of PACNS associated with ICH is different from that of PACNS associated with other neuroimaging patterns. Moreover, the difference between PACNS-related ICH and non-PACNS-related ICH has not been ascertained.

The aim of this narrative review is to describe the pathophysiology and neuroimaging patterns of PACNS-related intracranial hemorrhagic manifestations. Gaps in available evidence and potential research strategies to fill them will also be outlined.

## 2. Diagnostic Challenges in PACNS

PACNS is a rare primary vasculitis involving only the arteries of the leptomeninges, brain and/or spinal cord without systemic disease [6]. Its incidence in the general population is estimated at around 2.4 cases per 1,000,000 persons/year [7], with equal prevalence between genders and with a median age at diagnosis of around 50 years [8]. The currently used diagnostic criteria, as previously described, are summarized in Table 1. 

From the systematic evaluation of the disease categories emerging from the diagnostic criteria, it is highlighted that two different subtypes of disease are described on the basis of the most effective diagnostic technique, i.e., a subtype of angiography-proven PACNS and a subtype of biopsy-proven PACNS. These subtypes roughly correspond to large-vessel PACNS (LV-PACNS) and small-vessel PACNS (SV-PACNS), respectively. Indeed, the LV-PACNS subtype corresponds to the subtype diagnosable as a “probable” PACNS in the Birnbaum and Hellmann’s criteria [2], while SV-PACNS cannot be diagnosed without pathology according to the same criteria [2]. In fact, the caliber of the cerebral small vessels involved in the pathological process is such that they are essentially not visible with angiographic techniques, such as catheter angiography or Digital Subtraction Angiography (DSA), being ≤500 mcm [9]. In the subtyping of cerebral vessels by caliber, the category of medium-caliber vessels, being angiographically visible, currently falls within LV-PACNS. Accordingly, it is possible to diagnose a PACNS involving mainly the medium-caliber arteries as “probable PACNS” [2] by using DSA. It is possible that, in the future, the availability of more information on this subtype could allow it to be considered as distinct from the isolated involvement of the large vessels. The main findings in the two PACNS subtypes according to the diagnostic criteria are summarized in Figure 1. 

As highlighted in the literature analysis presented in the recent ESO guidelines on PACNS [5], the available papers have notable limitations and do not allow the majority of diagnostic questions to be answered in such a way as to support the criteria proposed so far. In particular, on the neuroradiological side, the description of neuroimaging patterns appears unsystematic, diverse and inconsistent from one study to another on both the vascular and parenchymal sides. This is a fundamental limitation that makes attempts at clinical, neuroradiological and pathological neurological correlation less precise in order to distinguish subgroups of patients who are homogeneous in some characteristics and to be able to define their natural history. The vascular imaging techniques presented in the available studies are also different in technological age and performance and in type. The performance of DSA is not comparable to that of Magnetic Resonance Angiography (MRA) and Computed Tomography Angiography (CTA) in terms of application on the basis of the diagnostic criteria for high-probability angiographic patterns [10]. Parenchymal imaging also suffers from the same variability and lack of systematization in the application of standardized reading and reporting criteria. 

These limitations, combined with the non-specificity of the clinical findings and those of the individual diagnostic techniques, make the diagnosis particularly challenging. Indeed, PACNS is an often-overlooked but probably dramatically underdiagnosed disease. This is valid in general and can also be applied to some subtypes of clinical presentation, including the acute intraparenchymal hemorrhagic pattern. In fact, a hemorrhagic presentation of PACNS, including both ICH and Subarachnoid Hemorrhage (SAH) patterns, was reported in only 13.6% of the patients in the studies selected to answer the PICO 2 in the ESO guidelines [5], combining old and new ICH and micro- and macrohemorrhages. Similarly, it was not possible to associate any neuroimaging pattern, including hemorrhagic, prominently with LV-PACNS rather than SV-PACNS. 

It is therefore unclear whether the presentation pattern with ICH has a different natural history from PACNS without ICH; whether it has a different natural history from other causes of spontaneous ICH; and whether, in patients with ICH, the possibility of underlying PACNS is insufficiently searched for. In fact, the main targets of diagnostic imaging in ICH patients are the large vessels with a suspicion of arteriovenous shunt and, on the opposite side, the small vessels with their parenchymal MRI markers as the main cause of spontaneous ICH.

## 3. Intracerebral Bleeding in PACNS 

### 3.1. Clinical Issues

Intracranial bleeding, in the forms of both ICH and SAH, has long been considered characteristic of some forms of RCVS, therefore more easily indicating the suspicion of RVCS than of PACNS, considering the supposed rarity of hemorrhagic presentation of PACNS. RCVS can be easily and reliably distinguished by the presence of recurrent thunderclap headaches or other clinical features [11,12]. Although brain hemorrhages can occur in both RCVS and PACNS [13,14,15], hemorrhagic presentation is frequent at onset in patients with RCVS, in the context of the dynamic balance between hypoperfusion and reperfusion/hyperperfusion. In PACNS patients, who have a longer prodromal period than RCVS patients, the timing and risk factors of ICH are not yet completely understood. Topcuoglu et al. [16] compared hemorrhagic and non-hemorrhagic PACNS in a retrospective, long-lasting single-center cohort of 49 patients (20 biopsy-proven) diagnosed from 1993 to 2015 and probed the mechanisms. In this cohort, only 10% of all PACNS cases had ICH, and often during the course of the disease rather than at the onset. Interestingly, in this small cohort, the occurrence of ICH as the first manifestation of PACNS was associated with exposure to sympathomimetic drugs. Therefore, the authors raised the hypothesis of a shared mechanism between PACNS and RCVS in ICH, proposing that sympathomimetic-drug-induced prolonged distal vasoconstriction might end in inflammation. This study has several limitations, partially accounted for by its retrospective design. In particular, 4/49 (8.2%) patients were included as having PACNS despite a negative angiogram and without pathological confirmation. The neuroimaging history of the patients was not investigated apart from hospital admission at the enrolling center, and hemorrhagic presentation of PACNS included microbleeds as well, if present on the initial brain imaging study. The 5/49 (10%) patients with hemorrhagic presentation did not differ from the non-hemorrhagic PACNS patients in age (mean age 52 ± 14 years vs. 51 ± 15 years; *p* = 0.917), and 4/5 (80%) were men. In 3/5 patients, the diagnosis was biopsy-proven, while 1 out of 5 patients had an inconclusive biopsy. The main vascular risk factors and the previous assumption of antithrombotic drugs were not significantly different in the two cohorts. Hemorrhagic presentation was associated with headache (100% vs. 43%, *p* = 0.022), particularly thunderclap headache (60% vs. 0%, *p* < 0.001), and sympathomimetic drug use (100% vs. 7%, *p* = 0.001). Interestingly, 1/5 of the hemorrhagic PACNS cases had illicit drug consumption as a trigger factor. In 4/5 of the included patients, the presentation was an acute lobar ICH (3 supratentorial and 1 cerebellar), and in 2/3 of cases of supratentorial ICH, satellite SAH was present. Only ¼ of the ICH patients had abnormal arterial pressure values at admission. Interestingly, it was hemorrhagic stroke that led to the diagnosis of PACNS in all 4 patients. In the first description of the Mayo Clinic cohort [14], 16/131 patients (12.2%) developed intracranial bleeding within 3 months after the diagnosis of PACNS (12 cases of ICH and 4 cases of SAH), but the rate of ICH at presentation was not reported. Similarly, in the French cohort [17] the timing of ICH was not provided in the 10/51 (20%) patients reported to have macrohemorrhages on MRI. Noticeably, this last finding is not equivalent to symptomatic ICH. 

In addition to the literature review presented by the recent guidelines [5], where the selection criteria for the available papers were rather rigorous to avoid the effect of single case reports and very small case series as well as that of qualitatively insufficient data, three previous systematic reviews [18,19,20,21] are available, with different objectives, largely focusing on the same papers despite having non-overlapping selection criteria. Krawczyk et al. [18] considered the clinical presentation of patients according to the biopsy- or angiography-proven PACNS subtype but did not provide any information about the clinical presentation of PACNS-related ICH. Similarly, McVerry et al. [19] and Beuker et al. [20] did not address this issue in patients with hemorrhagic presentation. The information available in the literature on this aspect is therefore very sparse and does not allow us to identify a peculiar clinical presentation of PACNS-related ICH or to draw conclusions about the presence of associated elements that can support or guide the clinical etiological suspicion. 

### 3.2. Neuroimaging Issues

In McVerry et al. [19], hemorrhage of any kind was reported as present on MRI in 76 patients (10.8% of all PACNS patients and 14.0% of all PACNS patients undergoing MRI). In Krawczyk et al. [18], ICH was present in 66/460 (14%) PACNS patients, with similar prevalence in both biopsy- and angiography-proven PACNS. In Sarti et al. [21], a parenchymal or subarachnoid hemorrhage was reported in 78/380 (20.5%) PACNS patients on MRI without distinguishing old from new ICH or ICH alone from SAH alone. In the same review, aiming to elaborate a clinical and neuroradiological flowchart to raise and substantiate the suspicion of PACNS, hemorrhagic pattern was included among the “minor neuroradiological features”. Similarly, there is no information in the literature relating to the location of the ICH in patients with PACNS and hemorrhagic presentation (e.g., deep vs. lobar, single vs. multiple, supratentorial vs. infratentorial) or the neuroimaging pattern correlated with a coexisting or intrinsic microangiopathy, as would be expected in the etiological diagnostic work-up of a patient with ICH. Thaler et al. [22], in a modern approach to vascular neuroimaging issues in a small cohort of PACNS patients, described 11/33 (33.3%) patients as having hemorrhage on MRI, without significant differences between SV- and LV-PACNS (*p* = 0.687), but 9/18 (50%) of LV-PACNS patients had a positive biopsy, implying that medium vessels were probably involved more than large vessels, with an overlap between medium- and small-vessel involvement. In fact, in the same cohort, only 11/25 (44%) of patients categorized as having LV-PACNS underwent DSA. Unfortunately, only 4/11 (36.4%) patients had ICH, but 4/11 (36.4%) had SAH and 4/11 (36.4%) had microbleeds, with a single overlap of the two findings in the same patient (SAH and microbleeds). In the subgroup of 12/131 patients from the Mayo Clinic cohort [14], the ICH location was frontal in 5 patients, parietooccipital in 2 patients, parietal in 2 patients, temporal in 1 patient, occipital in 1 patient and bihemispheric in 1 patient. In the small cohort (4 patients) of hemorrhagic PACNS described by Topcuoglu et al. [16], more details are provided in the description of individual cases: Case A had a small right frontal cortical hemorrhage with subarachnoid extension and leptomeningeal enhancement over the right hemisphere on post-gadolinium T1-weighted sequences, persisting over the following 3 months together with accumulation of small acute infarcts and persistent leptomeningeal enhancement; CTA was normal, and the diagnosis was achieved by brain biopsy.Case B had a right cerebellar hemorrhage on anticoagulant therapy and underwent emergent posterior fossa decompression. In the early follow-up, a new left cerebellar hemorrhage and an acute left pontine infarction occurred; the diagnosis was provided by histopathologic examination of the surgically evacuated tissue.Case C had a right parietal lobar hemorrhage with subarachnoid extension, multiple small disseminated acute and subacute infarcts and small rounded sulcal hyperintensities on a fluid-attenuated inversion recovery sequence (dot sign); the diagnosis was provided by the angiographic pattern involving distal branch arteries.Case D had a right temporal lobe hemorrhage with small disseminated acute infarcts and chronic microbleeds; diagnosis was provided by angiography, and biopsy was negative.

All patients had several diseases, and two of them were chronic users of illicit drugs; these factors might have affected the MRI pattern, in particular for chronic SVD markers, and triggered the ICH. None of the presented patients had any sign of amyloid deposition on pathology. 

Detailed information on the neuroimaging pattern of ICH is not available in most of the available papers. In particular, it is not possible to define a preferential location (e.g., lobar vs. deep) of ICH or to define whether it is single or multiple or what the risk of recurrence of ICH is in these patients. Similarly, there is no systematic description of the associated (chronic) SVD pattern, including hemorrhagic and non-hemorrhagic MRI markers and standardized for clinical and research purposes, regardless of PACNS, in STRIVE 1.0 [23] and 2.0 [24]. The association of ICH with microbleeds and/or cortical superficial siderosis is therefore not known, except partially because in some case series ICH and microbleeds are reported interchangeably as hemorrhage. There is even less information on other neuroimaging markers, including enlarged perivascular spaces, white matter hyperintensities, old lacunae and small recent subcortical infarcts or cortical microinfarctions.

The majority of spontaneous ICH events, both lobar and deep, have SVD as the main cause, but in the etiological classifications of ICH, PACNS is essentially not contemplated, despite being included in the classification of SVD [25]. In particular, not even the most recent proposals for the phenotypic classification of ICH [26], largely supported by neuroimaging, mention PACNS as a cause of ICH. This issue accounts for the potential underestimation of this disease as a cause of ICH.

Sometimes PACNS-related ICH may not manifest with a hyperacute appearance on neuroimaging, as shown in Figure 2 and Figure 3. 

Even the subtype of SV-PACNS, described over 20 years ago as Amyloid-Beta-related Angiitis (ABRA), in whose histopathological definition both the deposition of fragments of amyloid beta and transmural inflammation are present and in whose neuroimaging characterization with MRI there are clearly evident microhemorrhagic markers, it does not appear to be strictly associated with ICH, unlike the SVD to which it explicitly refers, i.e., cerebral amyloid angiopathy (CAA). In the first systematic description by Scolding et al. [27], only 1 out of 9 patients had an ICH that was not present at onset. Conversely, all patients had tumefactive white matter involvement, often in more than one lobe, resembling the description of the more common CAA-related inflammation (CAA-ri) [28]. Interestingly, in the largest published cohort describing the natural history of CAA-ri [29], 33.6% of patients had a history of ICH at enrollment, 71.7% had probable CAA according to the Boston criteria and the OR for ICH development at 24 months was 17.1 (95% CI 8.5–32.7). These characteristics outline a different type of disease and make it probable that, in so-called ABRA, beta amyloid is not the most important immunogenic element in the pathogenesis but may sometimes be an innocent bystander. A further limitation derives from the fact that, despite the need for a histopathological diagnosis of ABRA, as of all SV-PACNS, the perception of a relatively benign course; the clinical and neuroimaging similarity to CAA-ri, which has validated clinical–neuroradiological vs. histopathological criteria; and the substantial correspondence of the therapeutic approach, tends to limit reliable descriptions of affected patient cohorts. From a histopathological point of view, the difference between the two conditions in the acute phase lies in the perivascular and not transmural localization of the inflammatory infiltrate in CAA-ri as well as in the histopathological signs of association with Alzheimer’s disease, sporadic and poorly characterizing in ABRA and much more frequent in CAA-ri [27]. 

### 3.3. Histopathological Issues

There is no evidence of the preferential association of a histopathological subtype of PACNS with hemorrhagic manifestations, and, surprisingly, the majority of cases described in detail in the literature had no histopathological signs of associated beta amyloid deposition. In the series presented by Topcuoglu et al. [16], brain biopsy showed granulomatous angiitis with eosinophils and necrotizing angiitis but no evidence for amyloid deposition in case A; marked multifocal vasocentric inflammatory lesions and features of necrotizing angiitis with no evidence of amyloid deposition in case B; lymphocytic and eosinophilic infiltration within and around the arterial walls with no evidence of amyloid deposition in case C. The biopsy finding of case D was normal, with a diagnosis provided by DSA, and biopsy was not performed in case E, showing a similar angiographic pattern. In the series from the Mayo Clinic cohort [14] only 4/16 hemorrhagic patients underwent biopsy, and amyloid beta deposition was described in two of them without information about the severity grade. One patient had a granulomatous pattern, and three had a necrotizing pattern. DSA was performed in 13/16 patients with intracranial hemorrhage, being reported as consistent with vasculitis in 100%. However, the proportion of positive angiograms was not significantly different between hemorrhagic and non-hemorrhagic patients, including mainly medium-vessel involvement in both subgroups, rated in the paper as “small vessel vasculitis”. This misreporting represents an example of the absence of standardization in the identification and description of findings in the majority of available studies, suffering from the absence of a dedicated neurovascular approach, both in the neurological and neuroradiological fields, and not corresponding to the standardized terminology based on the categories of vessel caliber currently used, with a correct interpretation of the performance of the diagnostic techniques in the different subgroups. Another aspect worth mentioning is that in patients with hemorrhagic presentation, obtaining a tissue sample is often contextual to a neurosurgical operation to evacuate the hematoma; therefore it is an open brain biopsy of clearly damaged tissue. This, on the one hand, makes some diagnoses of PACNS substantially unexpected or not formally suspected and, on the other hand, provides samples that are sometimes of limited technical quality from which to define the histopathology. In patients with well-founded clinical suspicion and neuroimaging suggestive of large vessel involvement, biopsy samples are acquired more rarely unless neurosurgical treatment is needed, and so it is not possible to define whether there is an association or overlap between LV-PACNS and SV-PACNS.

ABRA is almost never associated with ICH, despite a histopathology in which a considerable severity of involvement of the cerebral vessels by the underlying CAA is also described [27]. The nine cases described by Scolding et al. [27] showed the presence of severe leptomeningeal and parenchymal CAA and mild to moderate chronic inflammation within the leptomeninges and in and around the walls of many amyloid-laden blood vessels. The perivascular and intramural inflammatory infiltrate consisted of lymphocytes, macrophages and few multinucleated giant cells.

## 4. Management Issues

Patients with PACNS-related ICH present not only important diagnostic challenges but also currently unanswered questions regarding therapeutic management. In particular, the most critical subgroup of patients is that with ICH related to LV-PACNS, because the coexistence of stenosis and/or occlusions of the large intracranial vessels makes the simultaneous coexistence and/or the sequential occurrence of ischemic and hemorrhagic manifestations more likely. There are no data in the literature that make it possible to identify a response to treatment and a natural history specific to PACNS-related ICH. Furthermore, no information is available that could allow one immunosuppressive therapy strategy to be favored over the other in these patients. On the other hand, as highlighted in the recent ESO guidelines on PACNS [5], data on treatment are fragmented and inconsistent, both for the management of the acute phase and for maintenance therapy. The quality of the data does not allow us to define whether the addition of immunosuppressive therapy to steroid therapy in the acute phase can improve the outcome, nor does it allow us to define the best maintenance therapy and its duration. 

Finally, data about antithrombotic treatment and its safety profile are very scarce (PICO 14 of the ESO guidelines on PACNS) [5] and heterogeneous. Overall, 92/314 patients were reported to take antiplatelets (mainly aspirin) in the selected cohorts, without clear differentiation between SV- and LV-PACNS. In the unique historical cohort partially reporting the efficacy and safety of aspirin [30], there was no statistically significant difference in the prevalence of intracranial bleeding between patients not taking aspirin and patients taking aspirin (6.5% vs. 13%).

## 5. Critical Appraisal and Future Perspectives

Intracranial bleeding in patients with PACNS appears to be a widely underestimated and overlooked but not rare manifestation in its phenotype and pathophysiological mechanisms, as well as in its clinical, neuroradiological and natural history correlates. Many aspects still need to be better defined and understood. In particular, the relationship with the subtype of PACNS (e.g., LV- vs. SV-PACNS) is not reliably defined, and so it is not clear whether it is a manifestation more likely associated with microangiopathic rather than macroangiopathic involvement. Its histopathological characterization is very rare, and even the initial proposals of association with a necrotizing pattern have not been confirmed. The association with PACNS and amyloid beta deposition is also very limited; this issue increases the uncertainty about the mechanisms underlying intracranial bleeding in patients with PACNS. As for all manifestations of PACNS, the neuroimaging description appears not to be standardized, and this makes most of the available data not comparable with each other. However, these latter largely rely on retrospective case series, which makes it more difficult to draw sufficiently strong conclusions.

The role of associated and independent risk factors has never been extensively studied in patients with PACNS. For example, there is no information available on the methods and effectiveness of controlling arterial hypertension, just as there is often a lack of information on therapies previously taken. Some of the cases described by Topcuoglu [16] are in themselves borderline cases, in which the intake of sympathomimetic drugs for a long period (for years in one patient) and the concomitant intake of illicit drugs may have played a role, if not as an etiological factor, then at least as a trigger of the hemorrhagic event.

The information does not improve starting from the opposite point of view, i.e., from a patient presenting with ICH, as in this case PACNS is not even contemplated as a possible cause in the etiological classifications. In a recently published series [31] of 40 patients undergoing brain biopsy during minimally invasive intracerebral hemorrhage clot evacuation, none of the patients in the sample showed vasculitic changes. 

The frequently long prodromal phase of PACNS, the age range at diagnosis/median age of 50 years), across the lifespan, and the possibility of having associated hemorrhagic risk factors make it more complex to identify red flags from which to suspect PACNS as the etiology of ICH when this is the first manifestation of presentation. Prospective, well-designed studies with a standardized management of neuroimaging studies in the investigation, interpretation and reporting methodology are therefore necessary; such studies can begin to fill the gaps in the current evidence available on this topic.

## 6. Conclusions

The hemorrhagic subtype of PACNS is not well understood at the moment, and the limitation of the available data might contribute to this result. Moreover, ICH is probably underreported, and it accounts for under 20% of published cases of PACNS. Usually, it is not reported as the first event in these patients, and the location is usually lobar, with a lack of preferential association with LV- or SV-PACNS. The available data do not make it possible to identify triggers or concomitant diseases as main contributors to ICH in PACNS patients. In addition, PACNS is not considered in the etiological classification of ICH. Therefore, as PACNS is not included in the diagnostic hypotheses of ICH, some cases might be missed. Future studies should address this issue with a prospective design and a standardization of neuroradiological techniques and neurovascular work-up in order to fill the gaps in the evidence. 

## Figures and Tables

**Figure 1 biomedicines-12-00459-f001:**
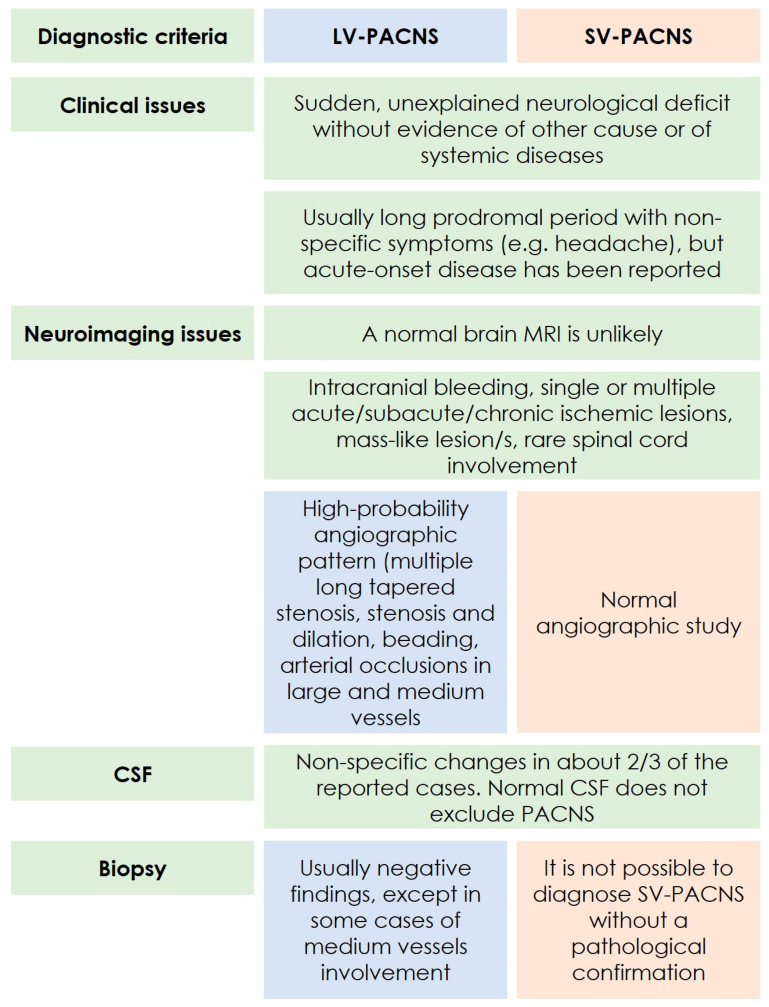
Schematic diagram of the main findings in PACNS based on the main points of the diagnostic criteria (see Table 1), considering LV- and SV-PACNS.

**Figure 2 biomedicines-12-00459-f002:**
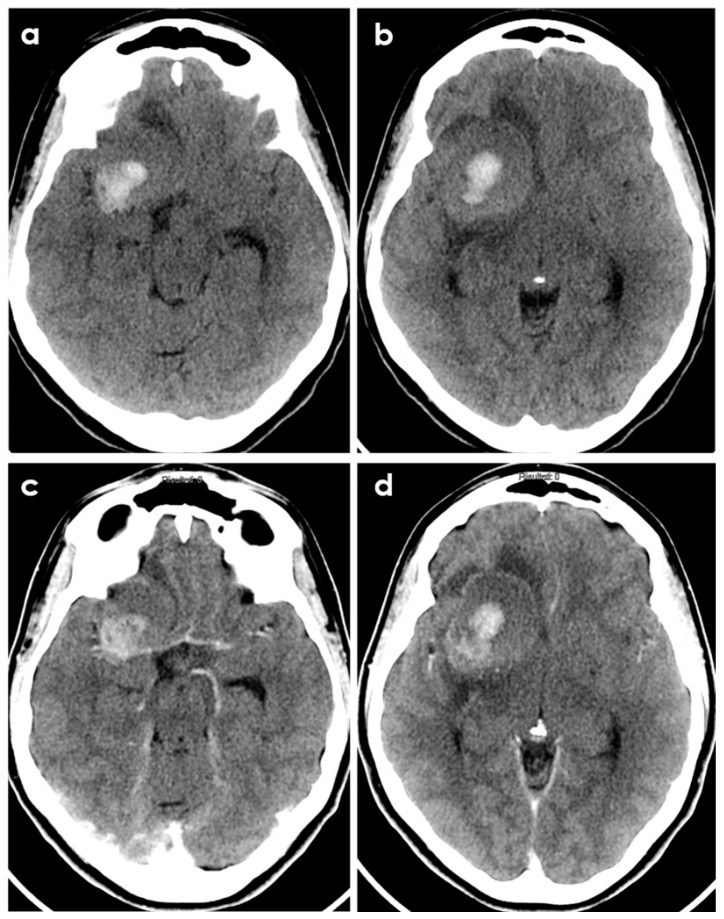
Brain CT of a young patient with biopsy-proven PACNS and ICH at presentation. The patient had a history of headache lasting 6 months before imaging. Panels (**a**,**b**) show the non-contrast CT appearance of a large rounded parenchymal hematoma with the different densities of the blood degradation products, surrounded by a hypodense rim of edema. Panels (**c**,**d**) show the corresponding post-contrast slices highlighting the displacement of the right MCA by the hematoma and the peripheral contrast enhancement of the lesion.

**Figure 3 biomedicines-12-00459-f003:**
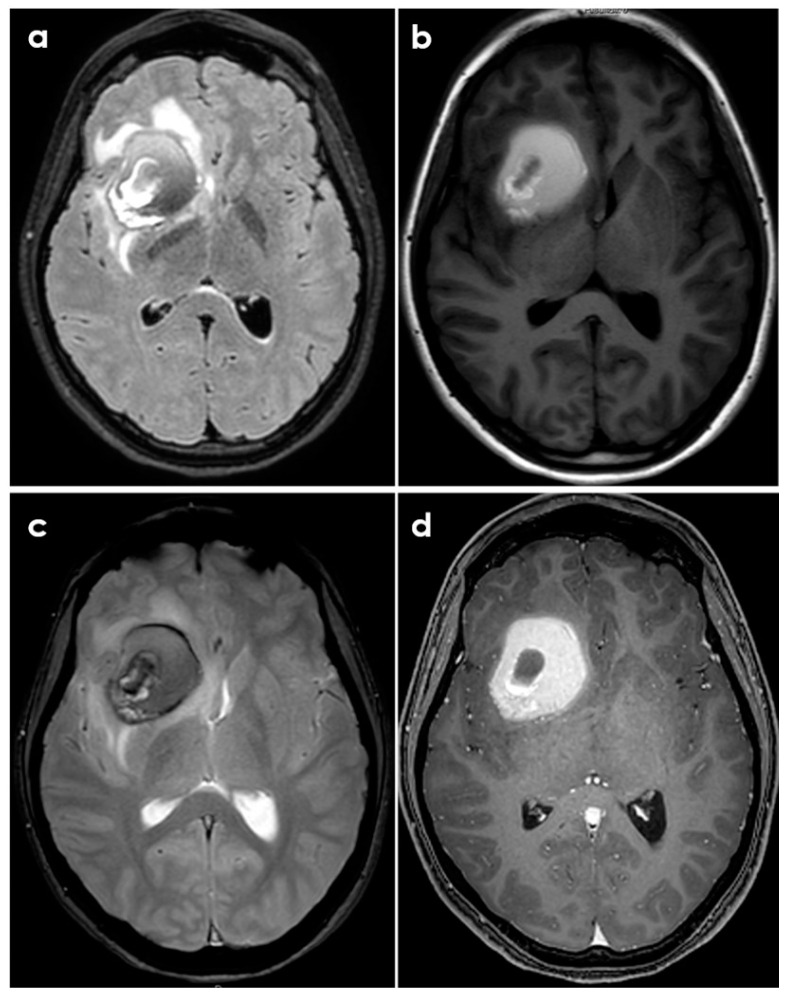
Brain MRI of the same patient described in Figure 2. Panel (**a**): axial fluid-attenuated inversion recovery (FLAIR) sequence, showing the rounded hematoma as a hypointense lesion and the surrounding edema as a hyperintense signal. Panel (**b**): T1-weighted axial sequence at the same level. Panel (**c**): gradient echo (GRE) sequence showing the various hypointensities of the hematoma components. Panel (**d**): post-contrast T1-weighted axial sequence showing the strong contrast enhancement of the hematoma in its peripheral ring.

**Table 1 biomedicines-12-00459-t001:** Diagnostic criteria of PACNS according to Calabrese and Mallek [1] and Birnbaum and Hellmann [2].

Year of Publication	Criteria
**1988** [1]	- A history or presence of a neurological deficit unexplained by any other cause after a thorough examination; - Evidence of vasculitis on either histopathology or angiography with changes characteristic of vasculitis; - Exclusion of a systemic vasculitis or any other condition to which the angiographic changes can be secondary
**2009** [2]	- A “definite” diagnosis of PACNS requires histopathological confirmation of vasculitis on cerebral biopsy or autopsy. - A “probable” diagnosis requires a high-probability angiogram with abnormal findings on magnetic resonance imaging (MRI) and a cerebrospinal fluid (CSF) profile consistent with PACNS.

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
