# Peer review of "The Hemorrhagic Side of Primary Angiitis of the Central Nervous System (PACNS)"

_biomedicines, 2024, doi:10.3390/biomedicines12020459_

Round 1

Reviewer 1 Report

Comments and Suggestions for Authors

I appreciate the authors for presenting this insightful review article. My comments are as follows:

1.  It is essential to include histology pictures to enhance the understanding of the content.

2. The management of Primary Angiitis of the Central Nervous System (PACNS) should be discussed.

3.  I recommend that the authors include a table or figure illustrating the guidelines for diagnosis or the pathophysiology of PACNS."

Author Response

I appreciate the authors for presenting this insightful review article. My comments are as follows:

We would like to thank the reviewer for his/her appreciation of our paper.

  1. It is essential to include histology pictures to enhance the understanding of the content.

Pathology is a fundamental step in diagnosing PACNS and in this comment the reviewer is right and we agree with him/her. Nevertheless, The main purpose of our review is to summarize lights and shadows (more shadows than lights) of hemorrhagic subtype of PACNS and not to highlight the diagnostic criteria of PACNS or address in detail vascular imaging and pathology. In fact, we are presenting a dedicated and expert point of view on the existing literature about this small subset of patients. Indeed, no correlation has been found between neuroradiological data and pathology and a single histhopatological pattern of hemorragic PACNS dioes not exist until now. Adding some pathological picture could reduce the attention of the reader on the focus of our review. Then, we respectfully prefer to do not add figures in this regard.  

  1. The management of Primary Angiitis of the Central Nervous System (PACNS) should be discussed.

It is not the focus of our review, as detailed in the introduction (“The aim of this narrative review was to describe the pathophysiology and neuroimaging patterns of PACNS-related intracranial hemorrhagic manifestations.”). We cited the recent ESO guidelines on PACNS and more than one of the authors of this paper contributed to the guideline production, so this reference can be proposed as updated view on PACNS management.  

  1. I recommend that the authors include a table or figure illustrating the guidelines for diagnosis or the pathophysiology of PACNS."

The diagnostic criteria are reported in table 1 and are criticized in section 2. As for the previous comment, we cited the ESO Guidelines (authored by three of the authors of the present paper) as updated and strong answer to this issue in order to keep the review short and easy to read. Pathophysiology is on of the most neglected and less understood fiwlds in PACNS, and hemorrhagic PACNS is even less understood. Then, we clearly expressed this view, proposing some pathophysiological issues from the literature in the description of triggers and neuroradiological issues.

Reviewer 2 Report

Comments and Suggestions for Authors

1) The abstract should clearly show what is new is being discussed in this review. What specific perspectives do the authors put forward after analyzing the array of information?

2) table 1 contains only 2 links. Has such a change in criteria really occurred over so many years? I believe that these statements should be supported by a large number of references to the literature.

3) This section should be illustrated with a block diagram.

4) Figure 1 and Figure 2 are their own images or borrowed. If the figures are borrowed, then the source must be provided.

5) The conclusion should be more meaningful and without abbreviations

Comments on the Quality of English Language

English quality is acceptable

Author Response

1) The abstract should clearly show what is new is being discussed in this review. What specific perspectives do the authors put forward after analyzing the array of information?

We would like to thank the reviewer for his/her comment. We better detailed in the abstract the aim of this review (“This review aims to summarize the available data addressing this issue with a dedicated focus on the clinical, neuroradiological and neuropathological perspectives. Moreover, the limitations of the actual data and the unanswered questions about hemorrhagic PACNS are proposed with the double point of view (PACNS subtyping and ICH etiology).”).

2) table 1 contains only 2 links. Has such a change in criteria really occurred over so many years? I believe that these statements should be supported by a large number of references to the literature.

We think that the reviewer catched a crucial point and this is exactly the message we are trying to propose. Validated and updated criteria for PACNS do not exist and the actual criteria (yes, only the two sets of criteria shown in table 1 exist) need to be updated. In this regard, we cited the ESO guidelines on PACNS, where thre of us are among the authors, and in this review we cited all relavant literature about the specific subset of PACNS of interest for the paper. This guideline contains “only” 76 references, considering all issues about PACNS (diagnosis and management), so the number of references of our actual paperi s proportionate to the topic (not PACNS, but hemorrhagic PACNS).

3) This section should be illustrated with a block diagram.

We understand the request of the reviewer, but we think that the answer to the previous comment can be extended to this one. Novertheless, if the Editor agrees, we confirm our willingness to add a diagram on PACNS diagnosis, but our fear is that this would distract attention to the main topic.

4) Figure 1 and Figure 2 are their own images or borrowed. If the figures are borrowed, then the source must be provided.

We confirm that the proposed images come from our personal case series and they were not previously published.

5) The conclusion should be more meaningful and without abbreviations

Many thanks for your comment. We changed the conclusions accordingly.

“The hemorrhagic subtype of PACNS is not well understood at the moment and the limitation of the available data might concur in this result. Moreover, ICH is probably un-derreported and it accounts for less than 20% of published cases of PACNS. Usually it is not reported as first event in these patients and the location is more often lobar with lack of preferential association with LV- or SV-PACNS. The available data do not allow to identify triggers or concomitant diseases as main contributors to ICH in PACNS patients. In addi-tion, PACNS is not considered in the etiological classification of ICH. Therefore, being PACNS not included in the diagnostic hypotheses of ICH, some cases might be missed. Future studies should address this issue with a prospective design and a standardization of neuroradiological techniques and neurovascular work-upin order to fill the gaps in the evidence.  “

Round 2

Reviewer 2 Report

Comments and Suggestions for Authors

the authors competently responded to my comments. the article may be accepted for publication in its current form

Author Response

Many thanks again to the reviewr and Editor. I uploaded the final version of the manuscript addressing the two points suggested by the Editor.